# Trends in per Capita Food and Protein Availability at the National Level of the Southeast Asian Countries: An Analysis of the FAO’s Food Balance Sheet Data from 1961 to 2018

**DOI:** 10.3390/nu14030603

**Published:** 2022-01-29

**Authors:** Syed Mahfuz Al Hasan, Jennifer Saulam, Fumiaki Mikami, Kanae Kanda, Nlandu Roger Ngatu, Hideto Yokoi, Tomohiro Hirao

**Affiliations:** 1Department of Public Health, Faculty of Medicine, Kagawa University, Kagawa 7610793, Japan; kanda.kanae@kagawa-u.ac.jp (K.K.); ngatu.nlandu@kagawa-u.ac.jp (N.R.N.); hirao.tomohiro@kagawa-u.ac.jp (T.H.); 2Department of Food Processing and Nutrition, Karnataka State Akkamahadevi Women’s University, Vijayapura 586108, Karnataka, India; jennifersaulam27@gmail.com; 3Department of Medical Informatics, Kagawa University Hospital, Kagawa 7610793, Japan; mikami.fumiaki@kagawa-u.ac.jp (F.M.); yokoi.hideto.md@kagawa-u.ac.jp (H.Y.)

**Keywords:** food, protein, food balance sheet, joinpoint regression analysis, jump model

## Abstract

We aimed to analyze the temporal trends in the per capita food (kcal/day/person) and protein (g/day/person) availability at the national level in the Southeast Asian (SEA) countries from 1961 to 2018. To avoid intercountry variations and errors, we used a dataset derived from the FAO’s old and new food balance sheets. We used the joinpoint model and the jump model to analyze the temporal trends. The annual percentage change (APC) was computed for each segment of the trends. Per capita food and protein availability in the SEA countries increased significantly by 0.8% per year (54.0%) and 1.1% per year (85.1%), respectively, from 1961 to 2018. During the 1960s, 1970s, and 1980s the per capita food availability in mainland SEA did not change significantly and was less than 2200 kcal/person/day. Since the early 1990s, food availability increased appreciably in the mainland SEA countries, except for Cambodia, which has experienced the increasing trend from the late 1990s. Distinct from the mainland, maritime SEA countries showed an up–down–up growth trend in their per-capita food availability from 1961 to 2018. Food-availability growth slowed down for Brunei (since the mid-1980s) and Malaysia (since mid-the 1990s) whereas it increased for Indonesia (1.5% per year), Timor-Leste (0.9% per year), and the Philippines (0.8% per year). Per capita protein availability trends in the mainland SEA countries were similar to the countries’ per capita food availability trends. Since the late 1980s, Thailand and since the late 1990s, other mainland SEA countries experienced a significant growth in their per capita protein availability. Since the late 1990s, per capita protein availability in Vietnam increased markedly and reached the highest available amount in the SEA region, following Brunei and Myanmar. Per capita protein availability increased almost continuously among the maritime SEA countries, except for Timor-Leste. Marked inequality did exist between maritime and mainland SEA countries in per capita food-availability growth till the mid-1990s. Considerable increases in per capita food availability have occurred in most of the SEA countries, but growth is inadequate for Timor-Leste and Cambodia.

## 1. Introduction

Food consumption is a key variable used to quantify and assess the developmental changes of the world’s food situation. The world has made significant improvements in raising the per capita food consumption [1]. This growth was accompanied by significant structural changes. The fast changes in dietary patterns might have been linked to the worldwide emergence of diet-related noncommunicable diseases [2]. Developing countries are going through a rapid nutrition transition coupled with the existence of the double and triple burden of disease [3]. The double burden of disease and changing dietary patterns has corresponded to nutrition transition in the Southeast Asian region [4].

Southeast Asia (SEA) is the home of about 8.6% of the world’s population, and accounted for 11.6% of the agricultural workers in the world [5]. This region alone contributes to about 7.7% and 15% of the agricultural production (gross production value) in the world and Asia [6]. Moreover, SEA provides about 35% and 47% of total rice exports in the world and in Asia, respectively [6]. The agricultural sector of SEA has increased its adoptions of modern technology since the 1980s, which resulted in steady growth, and it has continued to meet much of the food and nutritional demand within its regions [7]. An earlier report regarding the nutritional development in the SEA region reported that no country in the region had a food shortage at the national aggregate level and that all countries have attained food self-sufficiency—at least in terms of calories supply [8]. Per capita calorie and protein availability has increased steadily in the SEA region during the last few decades [7]. Calorie availability and protein availability have increased from 2194 kcal/day/person and 47 kg/year/person in the 1980s to 2686 kcal/day/person and 66 kg/year/person, respectively, in the 2010s. This growth was faster than in South Asia, while not as quick as in East Asia. Moreover, diversification in agriculture production resulted in the variations and changes in the compositions of protein intake in the SEA region, where both calories and protein have become increasingly available from animal sources [7]. Nutrition progress in SEA countries advanced together with the betterment of the food supply situation and economic development. These countries experienced a marked reduction in infant and under-five mortality rates during the 1960s, 1970s, and 1980s [8]. The prevalence of undernourishment has been reduced markedly in the SEA region (below 10%) except for Timor-Leste [9]. Though the SEA region has made remarkable progress in curbing the mortality rate among infants and children, it still has high and very-high prevalence of stunting in children under five years of age among most of the countries, except Singapore and Thailand [9]. On the other hand, prevalence of adult overweight and obesity has been increasing at a marked rate since the 1990s [9]. The rapid epidemiological transition has increased the probability of premature death from noncommunicable diseases in most of the SEA countries [10].

Adequate availability and constancy in the supply of food and, more prominently, access to food, are more important indicators of food security. Food availability at the national level has an important role to play in securing food security at the household level [11]. The gains in the per capita food consumption in the world reflected mainly those of the developing countries. The overall progress of the developing countries might have been influenced by the economic and political stability of these countries. Data on food availability at the national level of a country, in terms of kcal/person/day, provide important information and perceptiveness of dietary pattern and their sequential evaluation over time [12]. In this regard, the information in the SEA region in the literature has been somewhat piecemeal, focusing on short segments of periods with long discontinuity and on particular countries. Therefore, we aimed to analyze the temporal trends and significant changes in per capita food (kcal/day/person) and protein (g/day/person) availability in the SEA countries from 1961 to 2018.

## 2. Materials and Methods

### 2.1. Data Sources and Compilation

Similar to most of the countries, food supply data obtained from the food balance sheets is a comparatively dependable and conceivably the best available option to follow and evaluate the dietary transition at the national level. In this study, due to the lack of a long-term national dietary intake dataset, and to avoid intercountry variations and errors, food and nutrients availability data in the SEA countries were obtained from the FAO’s food balance sheets, which are compiled and documented in the Food and Agriculture Organization Corporate Statistical Database (FAOSTAT). We used the previous food balance sheet (FBS) data from 1961 to 2013, the last recorded years in the previous food balance sheets [13] and the updated food balance sheet (FBS) from 2014 to 2018, the last recorded years (during our data extraction period) in the updated FBS [14]. The updated FBS has made some methodological progression in calculations as compared to the previous FBS. The key distinction between the updated and previous FBS methodologies is the lack of a balancer variable such as stocks [15]. In the past, for example, stocks or feed components of the FBS caused almost all the statistical errors. In the updated FBS with the updated methodology, dedicated modules and a balancing mechanism are used to efficiently impute the FBS components and spread out the imbalances [16].

The food balance sheets for Southeast Asia (Total) and Southeast Asia (List) were downloaded as two separate comma separated values (.csv) files from the FAOSTAT database- one was from the previous FBS from 1961 to 2013 [13] and another one was from the updated FBS from 2014 to 2018 [14]. Afterward, we merged the two separate databases into one single database from 1961 to 2018. Neither the old nor the new FBS databases of FAOSTAT contain any food availability data on Singapore; hence we were unable to include Singapore in our analysis. In addition, in the new FBS database of the FAOSTAT, during our data extraction period (August 2021), there is no data available for Brunei; hence, we compiled the food and protein availability data of Brunei from 1961 to 2013. Per capita food and protein availability data are represented as kcal/day/person and g/day/person, respectively, in the FAOSTAT database [13,14].

### 2.2. Segments of the Southeast Asian Region

For this study, we compiled the data based on the FAO coding system for the countries of the world. Based on the FAO coding system, there are eleven countries in this region—Brunei Darussalam, Cambodia, the Lao People’s Democratic Republic (Laos), Malaysia, Myanmar, Philippines, Singapore, Thailand, Timor-Leste, and Vietnam. Southeast Asia (SEA) consists of the regions situated east of the Indian subcontinent and south of China [17]. Based on the contemporary definition, Southeast Asia consists of two geographic regions: mainland Southeast Asia and maritime Southeast Asia [17]. The phrase maritime Southeast Asia is used increasingly to refer to maritime aspects of the region’s history—a development linked with larger attention to transnational relations and to the sea itself as a part of history. Mainland Southeast Asia, also known historically as Indochina, comprises Vietnam, Laos, Cambodia, Thailand, and Myanmar. On the other hand, maritime Southeast Asia, also known historically as Nusantara, the East Indies, and the Malay Archipelago, comprises Indonesia, Malaysia, Singapore, the Philippines, Timor-Leste, and Brunei Darussalam [18]. To ease the description of our results, we adopted this division of the SEA region as mainland Southeast Asia and maritime Southeast Asia.

### 2.3. Trends Analysis

To analyze the trends and significant changes in per capita food and protein availability at the national level of the SEA countries, we performed a joinpoint regression analysis by using the Joinpoint Regression program (version 4.8.0.0, National Institute of Health, Bethesda, MD, USA) [18]. Joinpoint regression analysis is able to identify years when there was a significant change in the linear slope of the trends in food and protein availability from 1961 to 2018. The best-fitting points, called “joinpoints”, were identified when the rate changed significantly (*p* < 0.05) from 1961 to 2018. In each joinpoint, the trend changed its direction significantly in accordance with the transformation or changes in food and protein availability at the national level of the SEA countries. Therefore, each joinpoint acted as a changing point and illustrates the basis of the changes observed in food and protein availability. We hypothesized that economic development, technological advancement, policy-driven growth in agriculture, and changes in population growth might have acted as drivers [19,20] for these joinpoints, the nodes of changes in food and protein availability at the national level of the SEA countries.

The analysis began with a null model of no to a minimum number of joinpoints, and tested against whether the alternative model of one or more joinpoints (in this study up to five) was statistically significant. If the null model was rejected, the alternative model was selected and added to the better-fit model. We then computed the estimated annual percent change (APC) for each of those trends in food and protein availability. Furthermore, the average annual percent change (AAPC) was calculated as a geometric weighted average of APCs of various segments [20,21]. This AAPC value was used to summarize the total changes in per capita food and protein availability of each of the SEA countries over the analyzed period (1961–2018). We extended our analysis to an advanced level of the joinpoint regression model, the jump model, to estimate and adjust the effect of the introduction of the updated FBS on the underlying trends from 1961 to 2018. The jump model considered the introduction of the updated FBS since 2014, which caused a jump in the per capita values, but was assumed not to affect the underlying trends in per capita availability at the national level of the SEA countries [12,22].

## 3. Results

Per capita food availability in the SEA region significantly increased by 54.0% from 1961 (about 1836 kcal/day/person) to 2018 (about 2828 kcal/day/person), with an average growth of 0.8% per year. Protein availability (g/day/person) increased by 85.10% (AAPC = 1.1; average growth rate 1.1% per year) from 1961 to 2018 in the SEA region (Appendix A). In the two geographic regions of Southeast Asia, there were marked variations in food and protein availability. Average per capita food (50.1% vs. 40.9%) and protein (82.9% vs. 54.7%) availability increased more in mainland SEA compared to the maritime SEA countries, but there was a marked variation in the availability within the countries.

### 3.1. Trends in per Capita Food Availability in the SEA Countries

The trends in per capita food availability (kcal/day/person) in the SEA countries from 1961 to 2018 are presented in Table 1. From 1961 to 2018, the average growth in per capita food availability was found to be low in Cambodia (about 0.5% per year; only 30.0% increased) and Laos (about 0.5% per year; only 41.0% increased) whereas Myanmar showed the highest growth (about 1.1% per year; increased by 84.5%) followed by Vietnam (about 0.8% per year; increased by 59.0%) among the mainland SEA countries (Appendix A). In 1961, almost 1.7-fold differences were found in the per capita daily food availability among the SEA countries, with the highest availability being found in Malaysia (2418 kcal/day/person) and the lowest in Myanmar (1449 kcal/day/person). During the 1960s, 1970s, and 1980s the food availability in most of the mainland SEA countries did not change significantly and showed an almost sluggish changing trend, except in Cambodia and Thailand (Appendix A). During the first half of the 1970s, there was a marked decline in the food availability in Cambodia (APC = −6.85; decreased from 2016 kcal/day/person in 1972 to 1519 kcal/day/person in 1976). Thailand experienced significantly decreased food availability trends from 1978 to 1987 (decreased by 1.2% per year; from 2365 kcal/day/person in 1978 to 2068 kcal/day/person in 1987). During the 1960s, 1970s, and 1980s, per capita daily food availability in the mainland SEA countries was less than 2200 kcal. During this period, among the mainland SEA countries, Thailand had the highest food availability (average availability was around 2156 kcal/day/person) and Myanmar had the lowest (average availability was only 1542 kcal/day/person) compared to other countries in the mainland SEA. Food availability trends were found to be similar in Vietnam and Myanmar. During the second half of the 1980s, these two countries had experienced decreasing food availability trends. From the early 1990s, there was an appreciable increase in food availability in the mainland SEA countries—except in Laos and Cambodia, which experienced their increasing trends from the second half of the 1990s. The marked increasing trends of Cambodia slowed down (APC = 0.78) significantly from the early 2000s and those of Thailand slowed down (APC = 0.61) from the late 1990s. Since the early 1990s, per capita food availability in Vietnam increased noticeably by 1.83% per year and reached the highest availability in 2018 (3025 kcal/day/person) among the mainland SEA countries. Like Vietnam, Myanmar showed a promising trend in its per capita food availability from the early 1990s, which gained significant momentum during the 2000s (APC = 2.77; increased from 1975 kcal/day/person in 2001 to 2531 kcal/day/person in 2011) and since then slowed markedly (APC = 0.31). Since 1995, Laos experienced a marked increase in its per capita food availability (APC = 1.11) and from below 2000 kcal/day/person in 1995, calorie availability reached 2758 kcal/day/person in 2018.

During the 1960s, distinct from the mainland, maritime SEA countries experienced earlier increases in their per capita food availability except for Timor-Leste and Indonesia. Food availability was found to be the highest in Malaysia (2418 kcal/day/person) and the lowest in Timor-Leste (1741 kcal/day/person) and Indonesia (1824 kcal/day/person) in 1961 (Table 1). Average annual growth rate in per capita food availability were found to be low in Malaysia (AAPC = 0.4; only 18.0% increased) and Timor-Leste (AAPC = 0.5; only 31.0% increased), whereas Brunei showed the highest increase (AAPC = 1.2; increased by 55.0%) followed by Indonesia (AAPC = 0.9; increased by 58.0%) and the Philippines (AAPC = 0.8; increased by 49.0%) from 1961 to 2018. In Brunei, per capita food availability (Table 2) increased considerably since the mid-1970s by 2.9% per year until 1985 (from 2095 kcal/day/person in 1975 to 2902 kcal/day/person in 1985) and then slowed down noticeably to 0.23% per year until 2013 (2985 kcal/day/person). Food availability in Indonesia increased noticeably since the late 1960s by 1.6% per year until the late 1980s (from 1840 kcal/day/person in 1967 to 2588 kcal/day/person in 1988) and then, with a down–up–down trend remained almost same till 2004. Following 2004, Indonesia experienced a significant shift in its food availability trend with a 1.5% increasing rate per year (from 2471 kcal/day/person in 2004 to 2884 kcal/day/person in 2018). Starting from the highest per capita food availability among the SEA countries, Malaysia had an early, significant increasing trend until the early 1980s (APC = 0.69; increased from 2418 kcal/day/person in 1961 to 2766 kcal/day/person in 1983). Since then, the per capita food availability trend showed a zigzag pattern (Appendix A) with a marginal increase in the availability (only 3.0% increased from 1983 to 2018). During the second half of the 1970s, the Philippines experienced a notable momentum increase in its per capita food availability (APC = 2.71; increased from 1874 kcal/day/person in 1973 to 2273 kcal/day/person in 1980) followed by an interim decline during the early 1980s (about 10.0% decreased within 3 years). Since then, the Philippines had a fairly marked increasing trend in its per capita food availability (APC = 0.77; increased from 2043 kcal/day/person in 1983 to 2662 kcal/day/person in 2018). Similar to the Philippines, Timor-Leste had an amazing increase (about 4.8% per year) in per capita food availability during the second half of the 1970s, which then decreased significantly until the late 1990s (APC = −0.77; decreased by 13.4% and reduced to 1798 kcal/day/person in 1998). Since then, Timor-Leste had a fairly marked increasing trend in its per capita food availability (APC = 0.87; increased by 27.2% from 1998 to 2018).

### 3.2. Trends in per Capita Protein Availability in the SEA Countries

The temporal changes in per capita protein availability in the mainland SEA countries were similar to the countries’ food availability trends. In the mainland SEA countries, the average growth rate in protein availability (g/day/person) were found to be low in Thailand (about 0.6% increased per year; increased by 50.0%) and Laos (about 0.7% increased per year; increased by 56.0%) whereas like food availability trend; Myanmar showed the highest increase (about 1.4% increased per year; increased by 142.0%) followed by Vietnam (about 1.3% increased per year; increased by 117.0%) from 1961 to 2018 (Appendix A). Similar to food availability trends, during the 1960s, 1970s, and 1980s the protein availability in the mainland Southeast Asia did not change considerably and was almost unchanged except for Cambodia and Thailand (Appendix A). During the first half of the 1970s, there was a sharp and significant decline in per capita protein availability in Cambodia (APC = −9.09; decreased from 48.9 g/day/person in 1971 to 36.8 g/day/person in 1976) which then recovered by 1.95% increase per year from 1973 to 1991 (Table 2). Among the mainland SEA countries, protein-availability trends were found to be similar in Vietnam, Myanmar, and Laos. Plant protein contributed the bulk share of the available protein (Appendix A). Since the second half of the 1990s, there was an appreciable increase in per capita protein availability in the mainland SEA countries apart from Thailand which experienced an earlier increasing trend from the late 1980s. During this period, the animal protein share started to increase considerably (Appendix A).

The increasing availability trends in Thailand continued until 2009 and then slowed down significantly (APC = 0.84; only 9.8% increased from 2008 to 2018) and reached 63.4 g/day/person in 2018. Since the late 1990s, per capita protein availability in Vietnam increased appreciably by 3.20% per year and reached its highest availability in 2018 (98.6 g/day/person) among the mainland SEA countries (Table 2). Like Vietnam, Myanmar showed a potentially larger increasing trend since the late 1990s (APC = 4.75; increased by 1.7-fold from 47.2 g/day/person in 1999 to 82.2 g/day/person in 2011), which slowed down markedly during the 2010s (increased by only 10.3% from 2011 to 2018). During this period, both Myanmar and Vietnam experienced marked shifts in animal protein availability and decreased plant protein availability. Since the early 2000s, Laos experienced a significant increase in its per capita protein availability (APC = 1.25) and from 56.7 g/day/person in 2001 reached 76.9 g/day/person in 2018 (Table 2). Like Laos, Cambodia increased its per capita protein availability by 1.7% per year from 2001 (55.2 g/day/person) to 2018 (66.0 g/day/person).

In contract with the mainland, maritime SEA countries except Timor-Leste had experienced almost continuous increasing trends in their per capita protein availability from 1961 to 2018 (Appendix A). However, during the 2010s, mainland SEA countries had comparatively higher per capita protein availability compared to the maritime SEA countries (averaged 74.5 g/day/person vs. 70.1 g/day/person). Surprisingly, among the maritime SEA countries, protein availability was found to be the highest in Timor-Leste (averaged 63.1 g/day/person) and the lowest in Indonesia (averaged 35.5 g/day/person) followed by the Philippines (averaged 42.7 g/day/person) during the 1960s. During the 1960s, more than half of protein was contributed by animal sources in Timor-Leste, whereas in Indonesia about 85% of the available protein came from plant sources. From 1960s to mid-1980s, the protein availability trend in Timor-Leste was almost stable, and availability was found comparatively higher (averaged 62.7 g/day/person vs. 49.4 g/day/person) than other maritime SEA countries. Since 1984, per capita protein availability in Timor-Leste decreased by 1.5% per year till the late 1990s and then annually increased by 0.8% and reached 58.0 g/day/person in 2018 (Table 2). Like the food availability trend, per capita protein availability in Brunei increased considerably from 1975 to 1984 by 4.6% per year (from 53.5 g/day/person in 1975 to 87.1 g/day/person in 1985) and then slowed down noticeably to 0.24% per year and reached 93.4 g/day/person in 2013. Brunei showed the highest animal protein availability among the SEA countries from 1975 to 1984 (averaged 52% of the total protein availability) and after that Malaysia had the highest availability (averaged 55% from 1989 to 1993 and 56% from since 1993) until 2018 (Appendix A). In Indonesia, starting in 1963 availability increased noticeably by 1.5% per year until the mid-1990s (from 34.4 g/day/person in 1963 to 55.84 g/day/person in 1995), then reduced, then began to increase at a marked rate until 2018 (APC = 1.81; increased by 20.6%). Malaysia had a sharp significant increase (Appendix A) in protein availability during 1989 and 1993 by 5.4% per year (from 60.6 g/day/person in 1989 to 71.6 g/day/person in 1993) and then slowed down to 0.42% per year until 2018 (only increased by 8.0% in 25 years). Except for the period of 1982 to 1985, the Philippines experienced an almost steady increasing trend in its protein availability from 1961 (41.4 g/day/person) to 2018 (62.5 g/day/person). The Philippines had showed a stable share of plant (about 60%) and animal protein (about 40%) availability except for the period from 1984 to 1998 (Appendix A).

## 4. Discussion

In this multicountry ecological study, we analyzed the temporal trends and characterized the significant changes that have taken place in the per capita food and protein, availability at the national level of the SEA countries from 1961 to 2018. For the very first time, we analyzed the per capita food and protein availability at the national level of the SEA countries for this 58-year time span, by combining the FAO’s previous and updated FBS databases. This study provided the dynamics and temporal changes in per capita food and protein availability at the national level of the SEA countries in an omnibus way from 1961 to 2018 by using joinpoint regression analysis, and considering and adjusting the introduction of updated FBS data since 2014. In our analysis, we found that the introduction of the updated FBS significantly caused an increase or jump in food and protein availability in most of the SEA countries (data not shown in this article). Hence, trends in per capita food and protein availability at the national level of the SEA countries were analyzed on the basis of the jump model, where the effect of the introduction of the updated FBS since 2014 was considered [12] and assumed to not affect the underlying trends in food and protein availability. Above all, this is the first and the longest historical per capita food and protein availability data analysis of the SEA countries. Food and protein availability in the SEA countries increased significantly by 54.0% and 85.1% respectively, from 1961 to 2018. Average growth rate was 0.8% per year for food availability and 1.1% per year for protein in the SEA countries during 1961 and 2018. Per capita food (50.1% vs. 40.9%) and protein (82.9% vs. 54.7%) availability increased more in the mainland SEA compared to the maritime SEA countries, but there was a marked intercountry variation. Per capita food-availability growth rate was the highest for Brunei (1.2% per year) followed by Myanmar (1.1% per year) and the lowest for Malaysia (0.4% per year), followed by Cambodia, Laos, and Timor-Leste (0.5% per year). Protein availability at the national level increased more in Myanmar (1.4% per year) followed by Vietnam (1.3% per year) while it decreased for Timor-Leste (0.1% per year).

The growth in per capita food and protein availability at the national level of the SEA countries may result from the expansion of the agricultural sector. During the colonial era, the colonial powers invested little in improving the food production system of their colonies [23]. Most of the agricultural growth and development in the SEA region, and hence the increased availability in food and protein, occurred after the colonial era had been demolished. The unabated growth of the agriculture and aquaculture sector associated with the intensification of cultivation methods brought about significant yield increases for both cash as well as food crops [24]. From 1970 to 1995, cereal production increased by 117.8% in Southeast Asia. Cereal yield (tons/hector) increased by 65.6%, which resulted in 33.5% increase in energy consumption in Southeast Asia [23]. Moreover, the growth, development, and modernization of the agricultural sector of these countries [7] have contributed much to increase the per capita food and protein availability at the national level. Since the early 1960s, the SEA countries, particularly the national agricultural agencies of Indonesia, the Philippines and Malaysia, began to implement green revolution policies [25], first and foremost in rice agriculture, which raised the crop yields more and more rapidly. The green revolution policies increased the per capita food availability in the Philippines, Malaysia, and Indonesia during the 1960s and 1970s. Growth in per capita food availability in Malaysia was 0.7% per year during the 1960s and 1970s, whereas in the Philippines a remarkable growth (2.7% per year) was observed during the 1970s. Indonesia has experienced the benefit of the green revolution policies since the late 1960s. During 1970s and 1980s per capita food availability in Indonesia increased by 1.6% per year. During this period roughly about 50% to 66% of the calories from cereals and rice alone contributed more than 80% of the calories availability in Indonesia, the Philippines, and Malaysia (authors’ calculation from the FAO’s old food balance sheets).

During the 1960s and 1970s, the agricultural development in Malaysia, Philippines, Thailand, and Indonesia made food available above the subsistence level at reduced cost and facilitated the growth of the nonfarm economy, which eventually caused the agriculture-driven economic growth [26]. Vice-versa economic growth affecting the food availability at the national level, was observed in Malaysia, Indonesia, and Philippines. During the mid-1980s, Malaysia experienced a brief period of economic growth stagnation [11], which can be linked to the drop in food availability by 1.43% per year from 1983 to 1988. The Asian financial crisis caused a sharp contraction in the economy of Indonesia during the late 1990s [27], which may affect the reduced food and protein availability during that period. Moreover, during that time, there was a rapid increase in the price-level sharp depreciation in the Indonesian currency, which caused steep price inflation [27], Before that, during the late 1980s in Indonesia, a sharp decline in food availability was observed, which was due to the reduction in the supply of calories from cereal sources; the calorie supply from cereals reduced from 1708 kcal/day/capita in 1989 to 1557 kcal/day/capita in 1991. This reduction mainly occurred due to the reduced per capita availability of rice (reduced by 12.5% from 1989 to 1991; from 406.6 g/day/person to 355.6 g/day/person). The Philippines’ economy fluctuated widely during 1980s and registered a declined growth rate by 1.9% per year from 1980 to 1985 [11]. Political uncertainty, a mounting debt service burden and structural defects in the economy during the first half of the decade might have caused reduced food availability during the early 1980s. The results of food consumption surveys conducted in the Philippines in 1978, 1982, and 1987 indicated how food consumption reduced in the Filipino households when real per capita income declined [8]. Other than this, the slowdown in population growth in most of the SEA countries contributed to its overall increase in per capita food availability [11].

During the 1960s, 1970s, and 1980s the food availability in mainland Southeast Asia did not change drastically and was less than 2200 kcal/person/day. Since the early 1990s, there was an appreciable increase in the per capita food availability in the mainland SEA countries apart from Laos and Cambodia, which experienced the increasing trends since the late 1990s. The rhythm, as well as the magnitude, of the adoption of the green revolution was not uniform throughout the SEA region; some countries, mainly on the mainland, stayed behind for a while [28]. However, progressively all of the mainland SEA countries got involved in agricultural intensification. Since the mid-1980s, Vietnam remarkably reformed and implemented its economic reform policies in the context of so-called renewal or Doi Moi (Vietnamese language, meaning “innovate” or “renovate”) [28]. This green revolution allowed for a rapid growth in double annual cropping as well as an increase in yields per cropping season, which in turn led to an increased rice surplus in the SEA region, mainly to Thailand and Vietnam [28]. Since 1989, with international collaboration, Cambodia released 35 well-adapted rice varieties and made a major development with the increase in areas of double cropping of rice in the 1990s [29]. Major developments in the irrigation system in the mountainous northern Laos increased its production of rice. As in Cambodia, dry-season irrigation and the practice of double-cropping increased the availability of rice in Laos [29] and hence the Calories availability. In Cambodia and Laos, more than about 80% of the calories from cereals and rice alone contributed more than 90% of the calories availability. If food adequacy is measured at 2300 kcal/day/person [11], all the SEA countries already have had adequate food at the national level. The adequacy in food supply at the national level, economic development, and increased access to primary health care have resulted in the nutrition progress in SEA countries. The prevalence of undernourishment has reduced markedly in the SEA region since 2001. Less than 10% of the people in this region are undernourished—with one exception in Timor-Leste, where, alarmingly, more than 40% people are undernourished [9]. In addition, FAO estimates that 4.8% of the region’s population experienced severe food insecurity and 18.6% had moderate food insecurity, which is much lower than in southern Asia, Africa, Latin America, and the Caribbean [9]. As the consequence of increased food availability, the prevalence of undernutrition started to fall in this region during the 1980s and the improvement was remarkable in Thailand and Indonesia [8]. SEA countries experienced a marked reduction in infant and under-five mortality rates during the 1960s, 1970s, and 1980s [8]. The decline was particularly remarkable in Thailand, Malaysia, and Indonesia, whereas a moderate reduction was experienced by Myanmar, the Philippines, and Vietnam from 1960 to 1989 [8]. Though the SEA region has made remarkable progress in curbing the mortality rate among infants and children, however, still most of the SEA countries, except Singapore and Thailand had high and very-high prevalence of stunting among children under five years of age in 2019 [9].

In most of the SEA countries, except Brunei and Malaysia, bulk share of the available protein comes from plant sources, especially cereals (Appendix A). Since the mid-1980s, Brunei and Malaysia had more than 50% of their available protein from animal sources. The temporal changes in per capita protein availability in the mainland SEA countries were similar to the countries’ per capita food availability trends. Similar to food availability trends, during the 1960s, 1970s, and 1980s the protein availability in mainland Southeast Asia did not change considerably. In Southeast Asia, rising incomes, the growing population, and mounting urbanization have contributed to growth in livestock production and meat consumption, particularly poultry and pork [30]. In addition to livestock products, fish products are important in the diet of much of the SEA countries. The population usually has a high intake of fish. Fish is also a major source of animal protein in this region. Total fisheries production (both aquaculture and the capture fisheries) has more than tripled from 1980 to 2006 in the SEA region. The capture fisheries remain the chief source of fish production, but an emergent source of fish comes from aquaculture. Moreover, the share of aquaculture in total fisheries production in the SEA region increased from 10% of total in 1980, to 17% in 2000, and to 27% in 2006. By 2006, therefore, more than a quarter of total production of food fish came from aquaculture [31]. Since the second half of the 1990s, there was a significant increase in protein availability in the mainland SEA countries—except for Thailand, which showed an earlier increasing trend from the late 1980s. The increased protein availability in this region is mainly contributed by animal sources. Since the late 1990s, per capita protein availability in Vietnam increased appreciably and reached the highest available amount among the SEA countries, followed by Brunei and Myanmar. In contrast with the mainland, maritime SEA countries apart from Timor-Leste had experienced almost continuous increasing trends in their per capita protein availability from 1961 to 2018. Economic development increased per capita food availability and the shift from plant-based to more animal-based food might be linked to the increased overweight and obesity trend in the SEA region. The prevalence of adult overweight and obesity has been increasing at a marked rate since the 1990s [9]. Among the SEA countries, Malaysia has the highest prevalence (more than 40%) of overweight and obesity followed by Brunei [9]. There is a marked shift in the prevalence observed since 2000 among all the SEA countries. The rapid epidemiological transition has increased the probability of premature death from noncommunicable diseases in most of the SEA countries [10]. Consistent with the rise in overweight and obesity, diabetes has been rising gradually in this region [7]. In Malaysia, Brunei, and Singapore more than 10% of the adults suffer from diabetes [7].

The FAO’s food balance sheet data have some limitations that need to be addressed. Per capita food and protein availability data derived from this source do not provide the actual intakes of calories and protein at the household and individual levels of the SEA countries. Moreover, from this database we could not address the regional differences within a country. This is an average tentative quantity of food and protein available for consumption at the household or individual level. For that reason, it is impractical to make an inference and generalize the findings to the individual or subnational level of the SEA countries. There is a latent issue of overestimating per capita food availability because the FBS does not take into account food losses at the retail level. However, the FBS is the only cost-effective tool for analyzing the long-term changes as well as long-running comparisons of dietary pattern at a national level in the SEA countries. Studies indicate that food balance sheets are valuable for trend analysis as the databases are standardized and updated regularly [32]. Additionally, the use of FBS data to analyze the trends in nutrient availability in a country is more reliable than using absolute values at a single point in time [32].

## 5. Conclusions

In this study, we revealed the temporal trends in per capita food and protein availability at the national level of the Southeast Asian countries from 1961 to 2018, the largest historical analysis of the per capita food and protein availability at the national level of the Southeast Asian countries to date. We found that among the SEA countries, growth in protein availability was higher than the growth in calorie availability but cereals contributed the bulk share of calories and protein availability among the countries except Brunei. During the 1960s, 1970s, and 1980s the food availability in the mainland SEA did not change notably and was less than 2200 kcal/person/day. Since the early 1990s, per capita food availability increased significantly in mainland SEA except Cambodia. Distinct from the mainland, maritime SEA countries showed an up–down–up growth trend in their per capita food availability from 1961 to 2018. Since the late 1980s, Thailand and since the late 1990s, other mainland SEA countries experienced a significant growth in their per capita protein availability. Since the late 1990s, per capita protein availability in Vietnam increased markedly and reached the highest available amount in the SEA region following Brunei and Myanmar. Protein availability increased almost continuously among the maritime SEA countries, excepting Timor-Leste. Summed up briefly, Southeast Asia, mainly mainland SEA defeated the substantive calorie deficit condition since the early 1990s. However, structural changes in food availability, such as reducing calories from cereals along with increasing protein availability, became features in the diet of most of the SEA countries. These changes in apparent dietary intake may act as the prime cause behind the reducing prevalence of malnutrition but also the increasing prevalence of overweight or obesity and the emergence of diet-related chronic diseases in this valued region.

## Figures and Tables

**Table 1 nutrients-14-00603-t001:** Trends in per capita food availability at the national level in the Southeast Asian countries ^1^.

	Trend 1		Trend 2		Trend 3		Trend 4		Trend 5		Trend 6	
	Period	APC ^2^	Period	APC ^2^	Period	APC ^2^	Period	APC ^2^	Period	APC ^2^	Period	APC ^2^
Maritime SEA												
Brunei ^3^	1961–1967	2.83 *	1967–1975	−0.83	1975–1985	2.89 *	1985–2013	0.23 *				
Indonesia	1961–1967	0.10	1967–1988	1.57 *	1988–1991	−2.67	1991–1995	1.82 *	1995–2004	−0.44 *	2004–2018	1.48 *
Malaysia	1961–1983	0.69 *	1983–1988	−1.43 *	1988–1996	1.62 *	1996–2003	−0.76 *	2003–2013	0.42 *		
Philippines	1961–1973	0.52 *	1973–1980	2.71 *	1980–1983	−2.87	1983–2018	0.77 *				
Timor-Leste	1961–1975	0.01	1975–1980	4.77 *	1980–1998	−0.77 *	1998–2018	0.87 *				
Mainland SEA												
Cambodia	1961–1972	0.88 *	1972–1976	−6.85 *	1976–1993	1.69 *	1993–1997	−2.23	1997–2002	3.64 *	2002–2018	0.78 *
Laos	1961–1966	1.06	1966–1985	−0.29 *	1985–1995	0.33	1995–2018	1.11 *				
Myanmar	1961–1973	−0.41	1973–1984	1.48 *	1984–1988	−1.85	1988–2001	1.61 *	2001–2011	2.77 *	2011–2018	0.31
Thailand	1961–1963	4.93	1963–1978	0.31 *	1978–1987	−1.17 *	1987–1997	2.35 *	1997–2018	0.61 *		
Vietnam	1961–1974	0.10	1974–1977	−2.24	1977–1986	1.37 *	1986–1991	−1.75	1991–2018	1.83 *		
SEA region ^4^	1961–1968	0.40 *	1968–1988	0.84 *	1988–1991	−1.17	1991–1995	1.79 *	1995–2001	0.30	2001–2018	1.14 *

^1^ Trends analysis detected joinpoints, which are points where the line segment of the per capita food availability (kcal/day/person) trends of the Southeast Asian countries are joined. Per capita food availability trends of the Southeast Asian countries were analyzed on the basis of the jump model, where the effect of the introduction of the updated food balance sheet since 2014 was considered. Each joinpoint denotes a statistically significant (*p* = 0.05) change in the trends of the food availability in the Southeast Asian countries from 1961 to 2018. ^2^ APC is the annual percent change within a trend in food availability; APC was calculated from the data-driven Bayesian Information Criterion (BIC) method for the joinpoint model. ^3^ Per capita food availability trends for Brunei Darussalam were analyzed from 1961 to 2013. The food availability data for Brunei Darussalam was not available in the FAOSTAT updated food balance sheet database from 2014 to 2018 when the data was analyzed. ^4^ SEA region—Southeast Asia region; food availability trends was analyzed from 1961 to 2018 for the SEA region. Food availability data in the SEA region were calculated from the entire ten-countries database from 1961 to 2013. From 2014 to 2018, the availability database in the SEA region did not include Brunei Darussalam, as the data was not available in the FAOSTAT food balance sheet database. * Denotes that the annual percent changes (APC) were significantly different from 0 for a specific trend (*p* < 0.05) in the per capita food availability trends of the Southeast Asian countries.

**Table 2 nutrients-14-00603-t002:** Trends in protein availability at the national level in the Southeast Asian countries ^1^.

	Trend 1		Trend 2		Trend 3		Trend 4		Trend 5		Trend 6	
	Period	APC ^2^	Period	APC ^2^	Period	APC ^2^	Period	APC ^2^	Period	APC ^2^	Period	APC ^2^
Maritime SEA												
Brunei ^3^	1961–1975	0.98 *	1975–1984	4.62 *	1984–2013	0.24 *						
Indonesia	1961–1963	−1.37	1963–1995	1.46 *	1995–2003	−0.51	2003–2018	1.81 *				
Malaysia	1961–1972	0.76 *	1972–1978	2.32 *	1978–1989	0.11	1989–1993	5.37 *	1993–2018	0.42 *		
Philippines	1961–1982	0.93 *	1982–1985	−1.87	1985–2018	0.95 *						
Timor-Leste	1961–1984	−0.01	1984–1998	−1.45 *	1998–2018	0.81 *						
Mainland SEA												
Cambodia	1961–1973	1.32 *	1973–1976	−9.09 *	1976–1991	1.95 *	1991–1997	−1.86	1997-2001	5.75 *	2001–2018	1.69 *
Laos	1961–1966	1.15	1966–1986	−0.54 *	1986–1998	0.74 *	1998–2001	4.32	2001–2018	1.25 *		
Myanmar	1961–1973	−0.19	1973–1984	1.47 *	1984–1989	−2.15 *	1989–1999	1.67 *	1999–2011	4.75 *	2011–2018	0.49
Thailand ^4^	1961–1968	2.37 *	1968–1988	−0.45 *	1988–1996	2.88 *	1996–2009	−0.09	2009–2018	0.84 *		
Vietnam	1961–1970	0.16	1970–1977	−1.19	1977–1986	0.99 *	1986–1991	−1.16	1991–1997	1.95 *	1997–2018	3.20 *
SEA region ^5^	1961–1991	0.71 *	1991–1995	2.25 *	1995–1999	−0.25	1999–2018	1.72 *				

^1^ Trends analysis identified joinpoints, which are points where the line segment of the per capita protein availability (g/day/person) trends of the Southeast Asian countries are joined. Protein availability trends of the Southeast Asian countries were analyzed on the basis of the jump model, where the effect of the introduction of the updated food balance sheet since 2014 was considered. Each joinpoint denotes a statistically significant (*p* = 0.05) change in the trends of the per capita protein availability trends of the Southeast Asian countries from 1961 to 2018. ^2^ APC is the annual percent change within a trend in protein availability; APC was calculated from the data-driven Bayesian Information Criterion (BIC) method for the joinpoint model. ^3^ Protein availability trends for Brunei Darussalam were analyzed from 1961 to 2013. The protein availability data for Brunei Darussalam were not available in the FAOSTAT updated food balance sheet database from 2014 to 2018 while the data was analyzed. ^4^ Per capita protein availability trend was analyzed based on the standard joinpoint regression analysis; the jump model was not considered for this analysis. ^5^ SEA region—Southeast Asia region; protein availability trends were analyzed from 1961 to 2018 for the SEA region. Protein availability data in the SEA region were calculated from the entire ten-countries database from 1961 to 2013. From 2014 to 2018, the per capita protein availability database in the SEA region did not include Brunei Darussalam as the data was not available in the FAOSTAT food balance sheet database. * Denotes that the annual percent changes (APC) were significantly different from 0 for a specific trend (*p* < 0.05) in the per capita protein availability trends of the Southeast Asian countries.

## Data Availability

Data supporting reported results can be found in the FAO’s food balance sheets documented in the Food and Agriculture Organization Corporate Statistical Database (FAOSTAT) at https://www.fao.org/faostat/en/#home (accessed on 7 August 2021).

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
