# Peer review of "Trends in per Capita Food and Protein Availability at the National Level of the Southeast Asian Countries: An Analysis of the FAO’s Food Balance Sheet Data from 1961 to 2018"

_nutrients, 2022, doi:10.3390/nu14030603_

Round 1

Reviewer 1 Report

It is a logical manuscript and quite conscientiously written.

I just have a few minor comments:

1.change literary expressions into scientific ones, incl. "Our intention", "agricultural realm", "lion share" (lion's)
2. is it not possible to show the results of the share of animal and plant proteins in individual regions? In lines 400-401 you wrote that "protein comes from vegetable sources, especialy cereals" - first, where does this information come from, and secondly, it should be a plant not vegetable
3. In my opinion, there is no justification - the nutritional context. I propose to add information on how the proportions of people with malnutrition and excess body weight changed in the analyzed period. In the conclusions (lines 460-46) you mentioned it, but in my opinion it should be discussed in more detail in the introduction or discussion

Author Response

Dear Sir/Madam, Please see the attachment

Reviewer 2 Report

Here Syed Mahfuz Al Hasan and colleagues analyzed the temporal trends in the per capita energy and protein availability at the national level in the Southeast Asian (SEA) countries from 1961 to 2018.

This work is informative that worth publishing within Nutrients readership and community. However, there are some minor concerns to be addressed before acceptance for publication.

To begin with, energy availability seems to be a specific term used in exercise science, which is the amount of dietary energy remaining after exercise, available for other physiological functions such as growth, muscle recovery and homeostasis. Therefore, the authors should notify the readers the specific meanings here referring to food availability or security.

Secondly, the authors should discuss necessarily the factors that can affect the food availability in SEA, like the economy improvement, the population growth, and here are some references to share:

https://www.fao.org/3/ab981e/ab981e0b.htm

https://www.unicef.org/rosa/reports/asia-and-pacific-regional-overview-food-security-and-nutrition

And to extend these factors and combine into the discussion section which will help the reader understand the observed trends more readily.

Finally, with the COVID-19 Pandemic ongoing at a unmanageable pace worldwide, how Coronavirus exacerbated international food insecurity, especially in families with children in SEA will be a hot topic deserving investigations in the future by the group of authors of this manuscript.

Author Response

(The authors gave the same response as above.)
